# Perceived access to PrEP as a critical step in engagement: A qualitative analysis and discrete choice experiment among young men who have sex with men

**Elizabeth A. Asiago-Reddy**[1]*, **John McPeak**[2], **Riccardo Scarpa**[3], **Amy Braksmajer**[4], **Nicola Ruszkowski**[5], **James McMahon**[6], **Andrew S. London**[2]

**1** Division of Infectious Disease, Department of Medicine, SUNY Upstate Medical University Hospital, Syracuse, New York, United States of America, **2** Maxwell School of Citizenship and Public Affairs, Syracuse University, Syracuse, New York, United States of America, **3** Waikato Management School, University of Waikato, Waikato, New Zealand, **4** Department of Sociology, SUNY Geneseo, Geneseo, New York, United States of America, **5** Division of Infectious Disease, Department of Pediatrics, SUNY Upstate Medical University Hospital, Syracuse, New York, United States of America, **6** University of Rochester School of Nursing, Rochester, New York, United States of America

\* reddye@upstate.edu

## Abstract

Young Men who have Sex with Men (MSM) continue to face disproportionate HIV risk. Despite its well accepted role in HIV prevention, pre-exposure prophylaxis (PrEP) uptake remains below desired goals. Systemic barriers to PrEP access, including insurance complexity, cost, and wait times to start PrEP may contribute to low PrEP engagement. We conducted in-depth interviews and designed a discrete choice experiment (DCE) to assess preferences for and barriers to PrEP access in the United States. **Methods**: We conducted in-depth interviews with 18 MSM aged 18–30 years old who were not on PrEP and created a DCE based on the results. For the DCE, a convenience sample of young MSM in the United States who reported recent condomless anal sex was recruited through social media applications. Consenting participants provided sociodemographic information and responded to a series of 10 choice tasks about PrEP access. Preferences were analyzed utilizing marginal willingness-to-pay (mWTP) methods. **Results**: In-depth interviews revealed preferences for highly effective PrEP and concerns about barriers to access due to insurance coverage and privacy. The online DCE was completed by 236 eligible MSM aged 18–30. The most-preferred PrEP package—with all elements significantly preferred over other options—was insurance covered, could be maintained confidential from parents and employers, was available immediately, and had an online option. Need to take out new insurance or add a supplemental insurance in order to cover PrEP significantly detracted from willingness to pay for a PrEP program. Attributes most associated with willingness to pay for PrEP were PrEP being covered by an insurance the client already has and insurance coverage that was private. **Conclusions**: Young MSM at high risk for HIV in the United States who are not currently on PrEP showed strong preferences for PrEP options that were covered by insurance and could be kept confidential from parents and employers. Lack of

**Data Availability Statement:** The vast majority of the data are available and have been included. The In-depth interview transcripts pertinent to this

manuscript were included, with redacted words to preserve anonymity of the participants.

**Funding:** EAR received P30AI078498 from the U.S. National Institutes of Health (NIH.gov). Other team members, AB, NR, AK, JOM received direct or subcontracted reimbursement from this grant for work on this research. JMM received salary support as co-director of the University of Rochester Center for AIDS Research, funded on P30 AI078498 (https://www.niaid.nih.gov/). The funders had no role in study design, data collection and analysis, decision to publish, or preparation of the manuscript.

**Competing interests:** The authors have declared that no competing interests exist.

these options may present major barriers to PrEP access among young MSM who are at particularly high risk. Rapid access to PrEP, as well as the option of receiving some care online, may also enhance PrEP uptake.

## Introduction

Young men who have sex with men (MSM) in the U.S. continue to face disproportionate HIV risk. Recent data demonstrate increasing HIV incidence in the 25–34 year age group and among Hispanic/Latino MSM, as well as continued exceptionally elevated infection rates among Black/African American MSM [1]. Pre-exposure prophylaxis (PrEP) for HIV is a highly effective option for HIV prevention that recently received a Grade A rating from the United States Preventive Services Task Force [2]. While the uptake of PrEP in the U.S. has steadily increased over the last several years, PrEP initiation and retention among those at highest risk remains below desired goals [3–6].

Structural barriers to PrEP access, including insurance coverage and access to appointments, likely contribute substantially to suboptimal engagement in PrEP among those most at risk. Qualitative interviews and online surveys reveal that a belief that PrEP is unaffordable or not insurance-covered deters MSM of all ages with high-risk behavior patterns from exploring PrEP options [7–9]. Even with incentives, uninsured individuals at high risk for HIV are less likely to engage in PrEP care [10], while suboptimal levels of and delays in use of PrEP may lead to new HIV infections [11]. The U.S. PrEP Demonstration Project enrolled MSM and transgender women at public sexually transmitted disease clinics into PrEP care and followed them two-year period. Despite successes in enrollment and retention, once the Demonstration Project was over, uninsured participants were significantly less likely to continue PrEP care [5]. In the U.S., young adults are the group most likely to be uninsured, at about 30%, compounding the insurance challenges faced by those most at risk for HIV [12].

Access barriers aside from insurance, such as lack of clinic availability, or wait times for appointments or PrEP initiation, may also hinder PrEP access. Access to affordable primary care was significantly associated with PrEP continuation for PrEP Demonstration Project participants after the initial study period was complete [5]. Mikati et al. showed that same-day provision of PrEP improved uptake in New York City [13]. Online PrEP services may improve client satisfaction and prevent clinic appointment delays, but data assessing the relative access, safety, and retention associated with this approach are limited. Finally, privacy concerns related to PrEP access may contribute to lower uptake, particularly among young MSM who are covered by parental insurance [9,14].

Given the diversity of factors that may act as barriers to accessing PrEP among young MSM, novel ways of understanding how youth and young adults who are at high risk for HIV make decisions about PrEP are needed. A discrete choice experiment (DCE) is a unique form of survey increasingly utilized to assess health-related decision making [15–18]. In a DCE, participants are shown a repeated series of two "packages" or sets of options for a product or service (i.e., a choice task). Characteristics of the product or service are varied for each choice task, and participants are asked to choose which package they prefer, or to decline to uptake either package. DCEs are well-suited to PrEP research as they allow ascertainment of the relative importance of specific features of PrEP, called attributes, and the threshold for trade-offs between them. Such attributes can include location of PrEP care, how expensive it is, what type of delivery mode is used (e.g. pill or injection). Analysis of the willingness to pay for certain PrEP attributes offers a quantitative metric for determining the degree of importance of

the studied attributes. To date, few studies have analyzed PrEP preferences with DCEs, and none have focused specifically on young MSM at high risk for HIV in the U.S [19–23].

In order to better explore the degree to which access barriers which are modifiable either by patient choice, targeted policies, or drug manufacturers impact PrEP uptake among young MSM, we conducted in-depth interviews with MSM aged 18–30, and then designed a set of choice tasks based on our review of the literature and the priorities identified by those participants. The choice tasks assessed the importance of cost, insurance coverage, insurance privacy, wait times to PrEP start, and possible online PrEP delivery. Herein, we describe the creation and results of a discrete choice experiment pertaining to PrEP access among young MSM, who meet national guidelines for PrEP, but report never having taken PrEP.

## Materials and methods

### In-depth interviews

From July, 2018 to January, 2019 we conducted in-depth interviews with 18 cisgender men aged 18–30 residing in two Upstate New York mid-sized cities (Syracuse or Rochester). Participants were recruited through locally targeted Facebook advertisements and flyers on college campuses, walk-in sexual health clinics, and in local LGBTQ-friendly establishments including an LGBTQ youth center. Eligibility included cis-male or trans-female gender identity, condomless sex with a man in the last 6 months (oral or anal not specified), and never having taken PrEP. Enrolment criteria initially included young MSM aged 13 and above as well as transgender women, but due to lack of enrollment for MSM aged 13-17-years and transgender women, future work focused on MSM aged 18–30 years. In-person, open-ended, semi-structured, conversational interviews lasted between one and two hours. Interviewers were graduate students in sociology at a local university, and all identified as gay men; one was Black. The interview guide included questions about participants' experiences as young MSM, HIV/PrEP knowledge and risk perception, reasons for not participating in PrEP programs, and key elements that would be important to them both in terms of accessing and using PrEP. Additionally, each participant ranked characteristics of PrEP from most to least important to them and provided an account of why they ranked the attributes as they did. These characteristics were chosen after literature review regarding PrEP access and continuation in young MSM, as well as on the basis of contributions from a multi-disciplinary group of providers and peers who meet regularly to discuss the PrEP programs in Central NY. This group included authors EAR and NR.

After completing an interview, the interviewer audio-recorded a field note in which they identified important topics covered in the interview and any emergent topics that struck them as novel. These field notes were reviewed regularly and the new ideas that emerged during the course of fieldwork informed subsequent interviews. For the analysis, we transcribed all interviews verbatim and analyzed data using a modified grounded theory approach [24]. Given the deductive approach to interview guide development, we expected to hear participants discuss certain topics during the interview and we coded accordingly. We read the verbatim transcript data multiple times to identify all references to access to PrEP and the relevant data for anticipated and unanticipated themes related to PrEP access. Ultimately, we allowed for inductive analysis to produce the kinds of insights that come from grounded theory approaches to qualitative data analysis, even though we initially used deductive logics to organize the project and develop the interview guide. This modified grounded theory approach, in which codes are derived both from the literature and from the data, is quite standard in the analysis of qualitative data [25,26]. Coding was done independently by two separate investigators (ER and EB) in NVivo 11 (QSR International). Codes that were identified and refined included HIV

knowledge, HIV risk perception, PrEP knowledge, attitudes towards PrEP, beliefs about PrEP efficacy, perceptions of cost, ability to access PrEP care, trust in PrEP and the medical establishment, concerns about PrEP side effects, disclosure of PrEP use, and convenience of PrEP. Rare disagreements in coding were resolved by AL. The dominant themes identified in the interviews were utilized to inform the design of the DCE.

## Creation of the discrete choice experiment

Based on the results of the in-depth interviews (described in results), two separate DCEs were created. One focused on preferences related to accessing PrEP and the other on the qualities of PrEP as a medication (e.g., efficacy, side effects) and modes for receiving it (e.g., pill, injection). The creation of the DCE, as well as the results of the DCE pertaining to PrEP access are described in this manuscript. A companion article will present the results of the DCE on PrEP medication options.

The DCE was designed using Sawtooth Software Lighthouse Studio 9.6.0 (Sequim, WA, USA). We utilized a traditional full-profile conjoint-based choice design with balanced overlap. Non-logical combinations were excluded.

We determined which attributes of PrEP to include based on our review of the literature and the dominant themes reported by the in-depth interview participants. Inclusion of a cost variable was critical to allow for the estimation of the relative valuation of other attributes. The amounts of the potential monthly out-of-pocket costs were chosen based on the recommendations of in-depth interview participants, with a ceiling of $200 per month. Above that threshold, in-depth interview participants universally reported that they would not be willing or able to pay. Originally, the survey was planned to target participants in Upstate NY, and therefore to specifically reflect the population originally studied. It was ultimately decided that a meaningful sample size would not be obtained from local recruitment only, and that broader recruitment would be needed. Additionally, it was determined that understanding access preferences for a broader group of potential PrEP users from areas with diverse PrEP programs would provide more useful information to those designing PrEP initiatives. While using a local sample to design a national survey is an acknowledged limitation, our parallel inclusion criteria for both surveys attempts to mitigate this limitation.

## Eligibility criteria for the online DCE

We recruited a convenience sample of eligible young MSM through advertisements on Facebook and Grindr. Facebook advertisements were shown only to subscribers whose residence was listed in areas with population size <250,000 according to the U.S. 2010 Census, in attempts to recruit a more geographically diverse sample that reflected the mid-sized city population of the in-depth interviews. However, Grindr advertisements were delivered nationally without regard to population size as such an option was not available on their platform. Other platforms were explored but were not receptive to advertising. Potential participants were screened through a series of questions to meet the following eligibility criteria: U.S. residence, self-reported age 18–30, male sex at birth, not known to be HIV-positive, never on PrEP, and condomless insertive or receptive anal sex with another man outside of a mutually monogamous relationship in the last 6 months. Eligibility criteria were not made available in advance of consent and enrollment so as to avoid participants intentionally "screening-in."

## Identification of untrustworthy online surveys

The following circumstances led us to designate a survey untrustworthy/unreliable: 1) surveys which took less than 5 minutes to complete; 2) surveys completed from the same internet IP

address completed after an ineligible survey was collected from that IP address; 3) surveys from the same IP address unless all respondents had distinct e-mail addresses and different sociodemographic characteristics; 4) surveys with the same e-mail address (only the first eligible survey from the same e-mail address was accepted); and 5) surveys in which the age and year of birth (the former asked at the beginning and the latter at the end of the questionnaire) did not match. We removed untrustworthy/unreliable surveys from the data set prior to analysis [27].

## Ethics considerations

The Research Subjects Review Board of the University of Rochester approved this study's methodology (#63899). All eligible in-depth interview participants signed informed consent forms and received a participant payment of $40. They were also eligible for reimbursement up to $10 for travel to the interview site. Online participants were asked to view an online informed consent form and provided signed consent via touchscreen or mouse. Online participants were provided a $30 Amazon gift card delivered to an e-mail address of their choice for one complete, validated survey.

## Administration of the discrete choice experiment

Young MSM who screened eligible and opted to review and sign the online consent form were first shown a brief information sheet. This information sheet included a paragraph about PrEP (what it is, what it is used for), and an explanation about what a DCE is and how to select a preferred option or opt to not choose either option. Participants were then shown 10 random task choices related to PrEP access and 12 random task choices related to PrEP modalities (to be described in a separate manuscript). After they completed all of the choice tasks, they completed a questionnaire that included questions about their sociodemographic characteristics, behavioral risks, PrEP knowledge, and perceived PrEP access.

An example of choice task exploring PrEP access is provided in Fig 1. The possible options that participants could have seen in their choice tasks are displayed in Table 1. The full

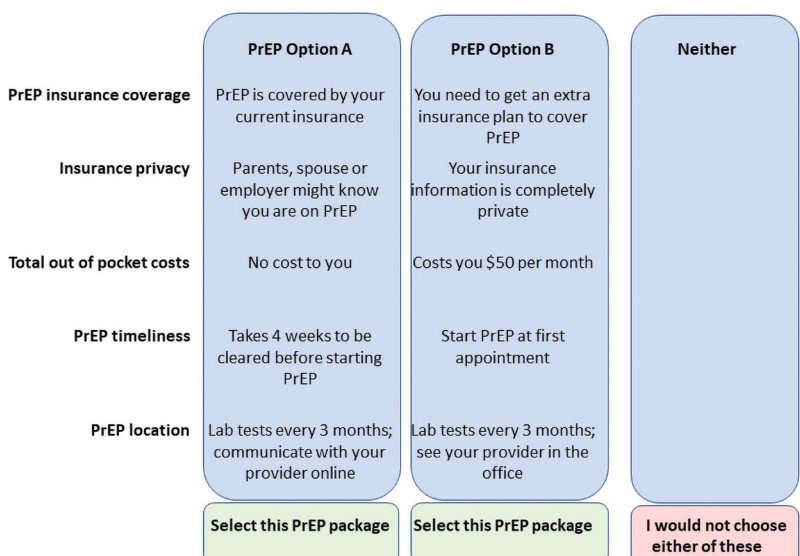

**Fig 1. Example of a choice task related to PrEP access**\*. \*This is one example only. Participants were shown a series of similar choice tasks with variation in the attributes presented; all possible attributes are listed in Table 1.

**Table 1. Possible attributes for PrEP choice tasks.**

| PrEP attribute | Baseline Option | Alternative 1 | Alternative 2 | Alternative 3 |
|---|---|---|---|---|
| **Total out of pocket costs** | Free | $20 per month | $50 per month | $200 per month |
| **PrEP insurance coverage** | PrEP is not covered by insurance | PrEP is covered by your current insurance | You need to get a new plan that covers PrEP | You need to get an extra insurance plan that covers PrEP |
| **Insurance privacy** | Parents, spouse, or employer might know you are on PrEP | Your insurance information is completely private | | |
| **PrEP timeliness** | Start PrEP at first appointment | Takes 1 week to be cleared before starting PrEP | Takes 4 weeks to be cleared before starting PrEP | |
| **PrEP location** | Lab tests every 3 months; see your provider in the office | Lab tests every 3 months; communicate with your provider online | | |

screening questionnaire, explanatory information, and close-ended questionnaire on sociodemographics and risks is available in the supplementary material, Appendix 1 in S3 File.

**Analysis of choice data.** Choice data were analyzed using a random utility mixed logit model for repeated choices by the same subject in R Studio Version 1.1.463 (The R foundation) [28,29]. This approach allows and accounts for heterogeneity of preference across subjects. To specify this model, let us denote with t the position in the sequence of stated choices by respondent n, with j the generic alternative in the choice set, and with i the selected alternative. Then, the random utility perceived from alternative i at the point t of the sequence by respondent n is denoted by an inner product between the k-dimensional vector of individual utility coefficients $\beta_n$ and the attributes of the alternative $x_{itn}$ to which is added a stochastic component of utility $\epsilon_{itn}$ known to the respondent, but unobservable by the researcher:

$$U_{itn} = \beta_n'x_{itn} + \epsilon_{itn} \tag{1}$$

This is the conventional random utility specified in "preference space." However, to ease the economic interpretation of estimated coefficients, utility was specified in money-metric space, rather than in the preference space, meaning that the degree of preference was demonstrated in a dollar amount [29]. Details on this specification are provided in Appendix 2. In our specification, we have normalized the estimation procedure with variation in the cost attribute. Our marginal willingness to pay (mWTP)-space approach makes the estimated parameters interpretable as representing the positive willingness to pay for a given level of an attribute compared to the baseline value, or if negative, how much a person would be willing to pay to have the baseline value compared to the specified attribute value. As the values for the cost of PrEP varied from $0 to $200 out of pocket costs per month, the monetary values of the mWTP reported in our results are interpretable as amount more or less a participant would be willing to pay for a given attribute compared to the baseline.

As described above, the estimation procedure provides us with a distribution of the mWTP values for a particular alternative value of an attribute in contrast to a baseline value of this attribute. This allows us to not only identify the average of this value in the population, but also provides an estimation of how these values are distributed around the average. Further, conditioning on the pattern of observed T choices in the sequence, we are able to derive the individual-specific means of mWTP for each respondent and to link its value to the characteristics of an individual as captured by responses to the background survey. This allows us to analyze PrEP access preferences for specific sub-groups within the overall sample. As noted in the opening section of this article, there is growing concern that sub-groups, such as Black and Hispanic/Latino young MSM, are experiencing elevated risks. This motivates us to further

investigate patterns within our estimated distribution of mWTP for different dimensions of PrEP access.

Critically, note that out-of-pocket costs from $0-$200 USD per month were always included as part of the packages that participants could select to allow us to estimate in mWTP space. Selecting a package with a particular dollar amount attached indicated a quantifiable willingness to pay for the remaining attributes of the selected package. A particularly attractive attribute of access to PrEP, therefore, would be expected to induce selection of a package even with a higher overall package cost.

In order to determine whether specific subgroups had preferences for PrEP uptake, independent variables were pre-selected based on epidemiological risk and being high priority for PrEP; given insurance concerns uncovered in our analyses preferences by baseline insurance were also explored. Subgroups investigated to determine whether unique preferences were observed were as follows: 1) Black race, 2) Hispanic/Latino ethnicity (race and ethnicity were not mutually exclusive), 3) High behavioral risk [participants who reported >10 anal sex partners in the last 6 months (highest quintile) and never using condoms], 4) Use of stimulant drugs in the last 6 months [30–32], 5) Self-reported high risk of contracting HIV (highest quintile), and 6) Insurance type.

In order to determine the influence of these variables on particular preferences for PrEP access, we specified a set of Ordinary Least Squares (OLS) equations in Stata 15.1 (StataCorp) [33]. We use the estimated means of mWTP values for each individual drawn from the probability density function (in USD) as the dependent variable. We stacked these OLS equations and estimated them as a system using a Seemingly Unrelated Regression (SUR) specification. We are able to recover the variance-covariance matrix of the estimated system and use this to develop joint tests of hypotheses concerning the statistical significance of a given independent variable across the system of equations. Note, however, given that the independent variables are identical in all 7 of the OLS regressions, we do not obtain any efficiency gain by using a SUR specification.

## Results

### In-depth interview results

18 in-depth interviews were conducted with MSM aged 18–26 (median, 20 years) who had never been on PrEP. Six were Black, 9 white and 5 of Hispanic/Latino ethnicity. All but one was a full or part-time student, and most were on parental (n = 9), public (n = 3) or student (n = 3) insurance. More details on this sample in are in Appendix 3. Interviews yielded three primary domains related to participants' ability to access PrEP. These included: 1) concerns about lack of insurance coverage and affordability; 2) concerns about privacy related to being on parental insurance; and 3) issues related to trust in the medical establishment, particularly around race.

**Insurance coverage/affordability.** The primary concern about accessing PrEP voiced by participants concerned insurance coverage and affordability. While most participants knew of a clinic location where they could access PrEP, many believed that they did not have the appropriate insurance coverage to pay for it. This belief served as a barrier to accessing PrEP even though there were mechanisms in place to mitigate that barrier. These interviews were conducted in New York State, which has some of the most comprehensive requirements for the inclusion of PrEP in insurance coverage that exist in the U.S., including a free public insurance option designated specifically for PrEP coverage. Despite the existence of these provisions in New York State, participants voiced concerns related to insurance coverage and affordability, and often linked affordability concerns to their decision to forgo PrEP. For example, one

20-year-old white man told the interviewer: "I don't know the exact cost. I know the medication is expensive. And it is the same expense—it is the same medication that someone who is HIV-positive would be taking. . .I just don't know the exact price but I know it is expensive and I know not all insurances cover it." Similarly, a 24-year old Black man told the interviewer: "I just feel like a lot of people in certain communities, especially the African American community, don't have the resources (to afford PrEP), unless they're of a certain class probably. I don't know. I just feel like all the resources aren't made available to everyone, basically."

Both of the participants quoted above were college students who had insurance through their university. Although they did not explicitly link such insurance coverage to PrEP access, other participants did. For example, one 25-year-old South Asian man, who had university-based insurance that had limited coverage told the interviewer: "I guess I didn't care [about getting on PrEP] because I don't have [good] insurance, so why bother knowing more about it? But yes, once I have an employment-based health insurance, I would like to look if they covered PrEP. But, because I didn't have [such an insurance], I just didn't even bother looking at it."

In addition to beliefs about lack of insurance coverage, concerns about affordability even with insurance emerged as a potential barrier to PrEP use. For example, one 22-year-old white man who was covered on a parent's insurance plan told the interviewer: "I went to the emergency room and [the insurance] sent a notice saying, you know, you can't use your insurance this much or we're gonna cancel your policy. And I had allergy shots and it was the same thing. . ..It seems very complicated and I don't want to run the risk of losing my insurance for my whole family. So better not."

**Insurance and parental privacy.** A second insurance-related theme, related to privacy, also emerged from our analysis of the in-depth interview data. Under the Affordable Care Act in the U.S., young adults can remain on their parents' insurance until age 26. However, private insurances will often provide explanation of benefit information to the primary person insured, thereby becoming a means of PrEP disclosure for youth and young adults on PrEP [34]. Concerns related to privacy and parental control of insurance emerged as important barrier to accessing PrEP for the young MSM with whom we spoke. For example, one 22-year old white man who was covered on a parent's insurance plan told the interviewer: "If the insurance company has this document that says I'm taking PrEP, and somehow that information is brought to my dad's knowledge, or brought to the knowledge of other people. . ..I would want to say no."

A secondary concern related to privacy emerged in relation to parental permission for youth who were younger than 18 years old, as exemplified by this conversation with an 18-year-old white man. This example reveals how stigma and privacy concerns, potentially related to being covered on a parent's health insurance plan, could converge to create a barrier to accessing PrEP for minors. [Participant]: "I was quite certain that I would not be—that there would be people who would look at me differently because I was a minor on a sexual prevention. It'd be like any girl who was on birth control before she turns 18. Like, 'Why do you need it? What makes you think you're going to need this before you turn 18?' [Interviewer]: "So before you were 18 did you think it would be hard to access PrEP?" [Participant]: "Mm-hmm. . ." [Interviewer]: "Did you know if you would need permission from a parent or guardian?" [Participant]: "I knew I would."

**Trust in medical systems.** In order for individuals to engage in PrEP care, a level of trust that PrEP may be of benefit to them needs to be in place. Features of trust in our sample centered around matters of race and racism in the U.S. For example, on 22-year-old black man told the interviewer: "It's just so crazy right now. Being black and that history, and things like that, it really messes with you. Being a young black man as well, it really makes you think like, 'Okay. Well,

who am I supposed to believe? Am I going to believe a conspiracy theory or facts?"' Similarly, a 24-year-old black man told the interviewer: "There's just not a lot of support to protect people's fears. And I'm hearing specifically from the Black community, because of African American history, like, 'I don't trust this government. I don't know what this pill's about.'"

**Prioritization of PrEP attributes.** Toward the end of the in-depth interview, after completing most of the open-ended questions, each participant was asked to complete a PrEP attribute ranking task. We gave each participant seven index cards. Each had an attribute, or characteristic of PrEP that might influence uptake, written on it (see Fig 2 for a list of the attributes and a pictoral of their rankings). The attribute displayed on the cards were chosen after discussion with key informants and stakeholders involved in the PrEP programs at the local hospital and health department. Individuals could add additional attributes that they thought important and missing in the set we provided. The interviewer then asked each participant to order the cards from most to least important to them, noting that there was no correct or expected result. The goal was to identify the participant's preferences. Once the participant was done, the interviewer recorded the order and talked with the participant about their thought process and how they assessed tradeoffs between the attributes.

As seen in Fig 2, effectiveness, cost and access were the highest ranked attributes, and the most likely to be ranked first by participants. Talking with participants about how they processed the bundle of attributes and ranked them was instructive. Concerns about PrEP affordability and ease of access were described as primary considerations before other PrEP attributes. For example, one 22-year-old white participant told the interviewer: "Well the cost for starters, if you can't afford it, you can't afford it. That's not really even a choice really if it's that expensive. Effectiveness, you know, if it's not reasonably effective, it's like what's the point." Similarly, an 18-year-old white man emphasized ease of access: "And to me, it's easy because if you can't get it, then its effectiveness doesn't matter. And if it's not effective you're not going to pay."

Based on the noted importance of access-related concerns in the in-depth interviews and the ranking exercises, two separate DCEs were created, one which specifically addressed concerns related to access to PrEP and a separate one which addressed PrEP modalities. For the access DCE, participants were asked to assume that the PrEP being offered was highly efficacious if taken correctly.

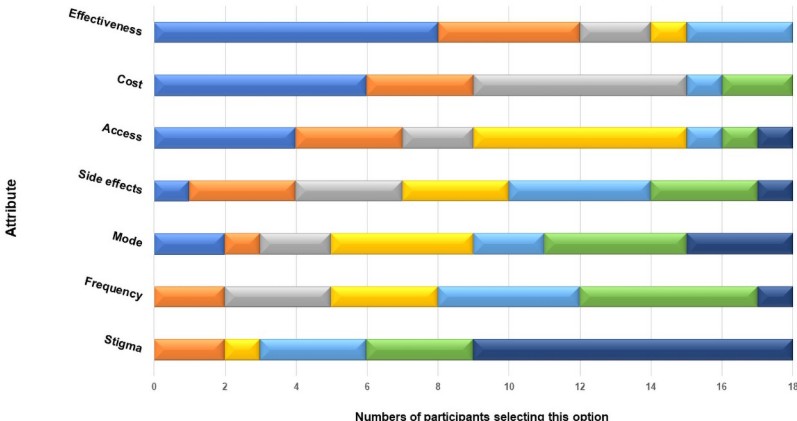

**Fig 2. Ranking of PrEP attributes among 18 young MSM during in-depth interviews in Upstate NY.** 1st Rank: Medium Blue. 2nd Rank: Orange. 3rd Rank: Gray. 4th Rank: Yellow. 5th rank: Light Blue. 6th rank: Green. 7th Rank: Dark Blue.

## Online survey results

Of 1297 online respondents, 882 screened ineligible or did not complete the eligibility questionnaire, 69 did not complete the online informed consent, 58 failed to respond to all of the questions, and 278 completed the study. Most ineligible respondents were excluded due to self-report of consistent condom use for anal sex (224), not having anal sex (115), or having one monogamous partner (158) in the last 6 months. From the 278 completed surveys, an additional 42 surveys were removed due to untrustworthy data (as described in methods), leaving 236 for analysis.

**Online survey sample description.** Characteristics of included participants from the national online survey are detailed in Tables 2–4. The median age was 24.5 years (interquartile range 21–26). There were 62 (26.3%) Hispanic participants, 36 (15.3%) black participants, and 5 (2.1%) who indicated both Hispanic and black (categories were not mutually exclusive). In contrast to the in-depth interview sample, which was limited to residents in of Syracuse and Rochester, NY, the online survey drew participants from all over the United States. About one-

**Table 2. Sociodemographic characteristics of 236 PrEP eligible young men who have sex with men from an online survey in the U.S.**

|  | Median | IQR |
|---|---|---|
| **Age** | 24.5 | 21–26 |
|  | N | % |
| **Race** |  |  |
| Asian | 17 | 7.2% |
| Black | 36 | 15.3% |
| Native American | 12 | 5.1% |
| White | 165 | 69.9% |
| Other | 15 | 6.4% |
| **Hispanic ethnicity** |  |  |
| Yes | 62 | 26.3% |
| No | 167 | 70.8% |
| **U.S. Census Region** |  |  |
| Northeast | 72 | 30.5% |
| South | 79 | 33.5% |
| Midwest | 35 | 14.8% |
| West | 62 | 26.3% |
| Pacific | 2 | 0.8% |
| **Student status** |  |  |
| Full or part-time student | 117 | 49.6% |
| Not a student | 116 | 49.2% |
| **Employment status** |  |  |
| At least part-time | 187 | 79.2% |
| Not employed | 43 | 18.2% |
| **Education** |  |  |
| Did not complete high school | 35 | 14.8% |
| Completed high school | 89 | 37.7% |
| Completed college | 111 | 47.0% |
| **Insurance coverage** |  |  |
| None | 34 | 14.4% |
| Parental | 82 | 34.7% |
| Public (e.g. Medicaid, Indian health Service) | 43 | 18.2% |
| Employer, exchange or other | 70 | 30.6% |

**Table 3. HIV risks of 236 PrEP eligible young men who have sex with men from an online survey in the U.S.**

| | N | % |
|---|---|---|
| **Condomless anal sex partners in last 6 months** | | |
| 1 | 36 | 15.3% |
| 2–10 | 161 | 68.2% |
| more than 10 | 35 | 14.8% |
| **Condom use during anal sex** | | |
| Never | 63 | 26.7% |
| Sometimes | 170 | 72.0% |
| **6 mos. female partners** | | |
| Yes | 67 | 28.4% |
| No | 169 | 71.6% |
| **Group sex in the last 6 months** | | |
| Yes | 75 | 31.8% |
| No | 159 | 67.4% |
| **Uses apps to find partners** | | |
| Yes | 219 | 92.8% |
| No | 16 | 6.8% |
| **Used the following substances in last 6 months:** | | |
| Marijuana | 128 | 54.2% |
| Other Stimulant/hallucinogen | 83 | 35.2% |
| Heroin | 2 | 0.8% |
| None | 97 | 41.1% |
| **Drinks >6 drinks/setting** | | |
| At least once a week | 47 | 19.9% |
| At least once a month | 65 | 27.5% |
| Less than once a month | 61 | 25.8% |
| Never | 61 | 25.8% |
| **Ever had** | | |
| Chlamydia | 64 | 27.1% |
| Gonorrhea | 61 | 25.8% |
| Syphilis | 26 | 11.0% |

third of the participants were respectively from the Northeast (30.5%) and South (33.5%), slightly fewer were from the West (27.1%) and Midwest (14.8%). About half were part- or full-time students (49.6%), slightly more than half had high school education or less (52.5%), and most were employed at least part-time (79.2%). About one-third of the participants had insurance through a parent (34.7%) or employer (30.6%); fewer had public insurance (18.6%), and 14.4% were uninsured.

In terms of sexual behavior and HIV risk, nearly all (219, 92.8%) used apps to find partners. Most reported 2–10 condomless anal sex partners in the last 6 months (161, 68.2%). About one-quarter (28.4%) reported having female as well as male sexual partners in the past six months, and approximately one-third (31.8%) had engaged in group or party sex in the last 6 months. More than one-quarter (63, 26.7%) reported never using condoms for anal sex, while the remainder reported using condoms sometimes; young MSM who reported always using condoms were not eligible for the study. Overall, 27.1% reported ever having chlamydia, 25.8% reported ever having gonorrhea, and 11.0% reported ever having syphilis.

Substance use was common among the young MSM who participated in the study. Over half reported marijuana use in the prior 6 months, and about one-third (35.2%) had used

**Table 4. HIV risk perception and PrEP perceptions of 236 PrEP eligible young men who have sex with men from an online survey in the U.S.**

| | N | % |
|---|---|---|
| **How worried about acquiring HIV in your lifetime?** | | |
| Not at all or a little worried | 92 | 39.0% |
| Moderately or very worried | 142 | 60.2% |
| **How would you rate your current risk of HIV?** | | |
| Low risk | 96 | 40.7% |
| Moderate risk | 92 | 39.0% |
| High risk | 45 | 19.1% |
| **How much PrEP knowledge do you have?** | | |
| Little knowledge | 50 | 21.2% |
| Some knowledge | 114 | 48.3% |
| Good knowledge | 70 | 29.7% |
| **How likely is it that you could access PrEP if you wanted it?** | | |
| Highly likely | 61 | 25.8% |
| Moderately likely | 119 | 50.4% |
| Unlikely | 28 | 11.9% |
| Impossible | 4 | 1.7% |
| Don't know | 22 | 9.3% |

stimulants, hallucinogens, or opioids (methamphetamine, cocaine, NMDA receptor agonists) in the past six months. Notably, very few reported heroin use. Binge drinking was common, with almost half reporting >6 drinks at one time in the prior 6 months. Almost 1 in 5 reported drinking that much weekly, including some young MSM who were under the age of 21.

Despite these demonstrable risks, the majority of participants considered their current risk of becoming infected with HIV to be low (96, 40.1%) or moderate (92, 38.9%). At the same time, 60.2% reported being moderately or very worried about acquiring HIV in their lifetime. While only 29.7% of participants reported good knowledge of PrEP, 76.2% said it was moderately or highly likely that they could access it if they wanted it.

## DCE results

Calculating willingness to pay for a particular PrEP option (all other options held equal) revealed statistically significant preferences for or against most of the alternative options as contrasted to the baseline options. Fig 3 demonstrates the vector size for each PrEP attribute displayed in the access DCE; those vectors which do not cross zero are significantly ($p < 0.05$) preferred over the baseline option, and vectors above zero generated a positive wiliness to pay, whereas those below zero demonstrated a negative willingness to pay and therefore a lack of desirability. The baseline option for insurance was "no insurance coverage." As Fig 3 demonstrates, insurance-covered PrEP was significantly preferred in comparison, and the need for a new or supplemental insurance to cover PrEP were significantly less preferred. Compared to insurance which was kept private from parents and employers, a lack of privacy was viewed negatively by participants, as was a requirement to wait to start PrEP after a baseline evaluation. The option to receive PrEP care online (with associated laboratory assessments) was also significantly preferred over standard in-person care. Summarizing these results, the most preferred PrEP access package would maintain privacy from employers and parents, would be insurance covered, would offer immediate PrEP initiation, and would have an option for online PrEP care.

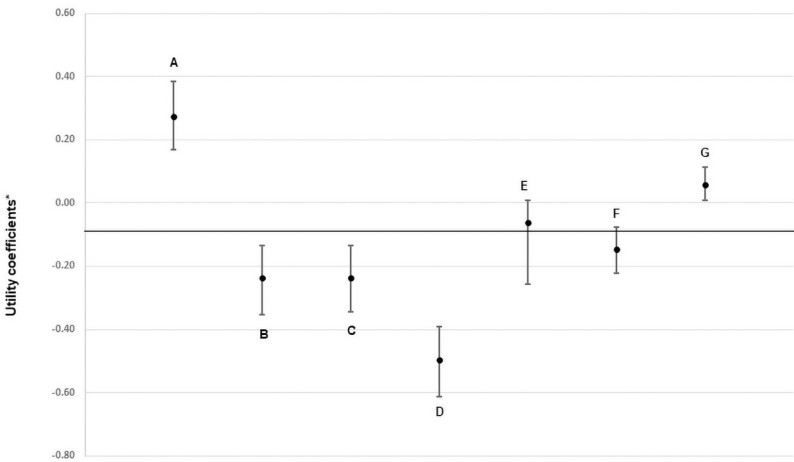

**Fig 3. Directionality of PrEP access preferences among 236 young men who have sex with men**[*]. [*] Negative numbers indicate an option that was less preferred than the baseline option. Attributes: A. Insurance covered, compared to not covered; B. Requires new insurance, compared to not covered; C. Requires extra (supplemental) insurance, compared to not covered; D. Insurance not private (services used can be seen by family or employers), compared to private; E. Wait 1 week to start PrEP, compared to immediate start; F. Wait 4 weeks to start PrEP, compared to immediate start; G. Communicate online with provider, compared to in-person visits.

A different pictorial view utilizing marginal willingness to pay estimates is shown in Figs 4 and 5. These figures demonstrate the proportion of participants who opted for a particular attribute at a given dollar value. The varied uptake at varied dollar levels creates a histogram with the mWTP corresponding to the peak of the histogram (smoothed to curves in the Figures). Curves which fall largely to the right of zero represent desirable attributes for which the participants are willing to pay more; the opposite holds true for curves falling to the left of zero. Curves with broader ranges demonstrate greater heterogeneity of preferences for a given attribute. For example, in Fig 4, insurance that is not private showed a negative willingness to pay, but a broad curve indicating a range of preferences for this attribute. As seen in Fig 5, an option for online PrEP was overall slightly but significantly preferred by almost all participants, with little heterogeneity (as exhibited by the narrow, high peak).

Breaking down willingness to pay numerically across the various attributes presented, a PrEP package that is not covered by a participant's current insurance, compared to a package that is insurance-covered, was worth $16.56 per month less on average. Requirements for new or supplemental insurance each result in significantly lower willingness to pay for a given PrEP package, -$69.93 for new and -$69.40 for supplemental insurance. This indicates that having no insurance coverage at all for PrEP was preferable to having to add a new insurance policy or to adding a supplemental insurance. Note that the distribution around the mean for new insurance or additional insurance is very similar in Fig 4, indicating that a requirement for a new or supplemental insurance were both seen as worse than not having insurance coverage at all. Overall, the value of privacy over not having this information private is $46.29 per month. As shown in Fig 5, there is an overall reduction in mWTP for having to wait for PrEP compared to immediate availability. Also, compared to immediate availability, a wait of one week was valued $6.19 per month less, while a wait of four weeks was valued $14.99 per month less. Finally, participants were willing to pay significantly more for an online PrEP package (with regular lab testing but no in-person visits) compared to a package that included standard in-person visits. The distribution of values for this access characteristic are relatively tightly distributed about the mean of $6.00 per month and all lie in the positive domain.

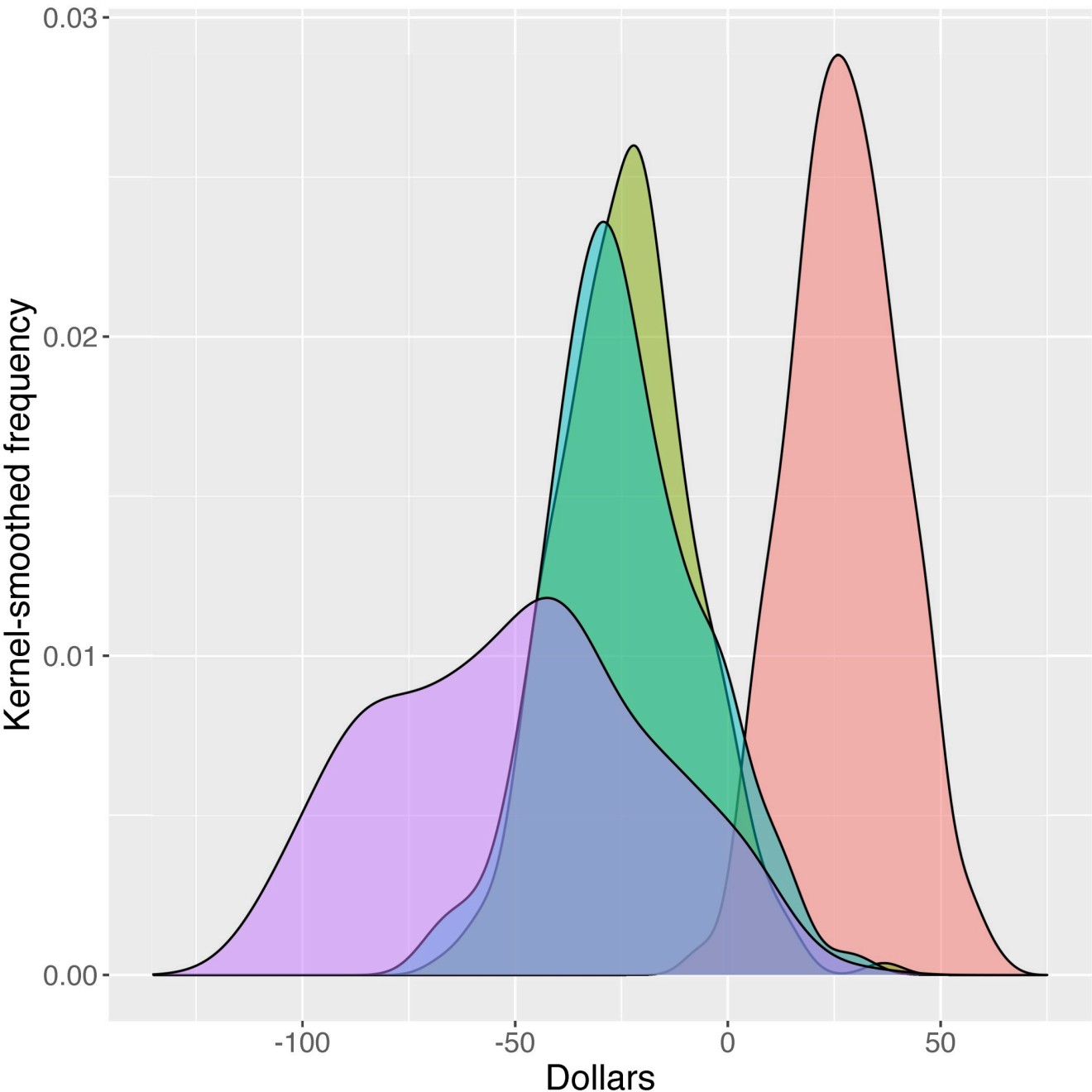

**Fig 4. Mean willingness to pay for PrEP among 236 young men who have sex with men in the U.S.: Insurance options.** Red: Insurance covered, compared to not covered. Green: Insurance covered, compared to not covered. Turquoise: Requires new insurance, compared to not covered. Purple: Insurance not private (services can be seen by family or employers), compared to private.

Overall, there were few significant differences between subgroups (by race or Hispanic ethnicity, objective HIV risk by number of partners or drug use, self-perceived risk, or insurance status). The most marked differences were noted in insurance preferences by insurance type (Fig 6). Participants who were on parental insurance were willing to pay significantly more for PrEP that was offered privacy from parents or employers. On the other hand, having to switch

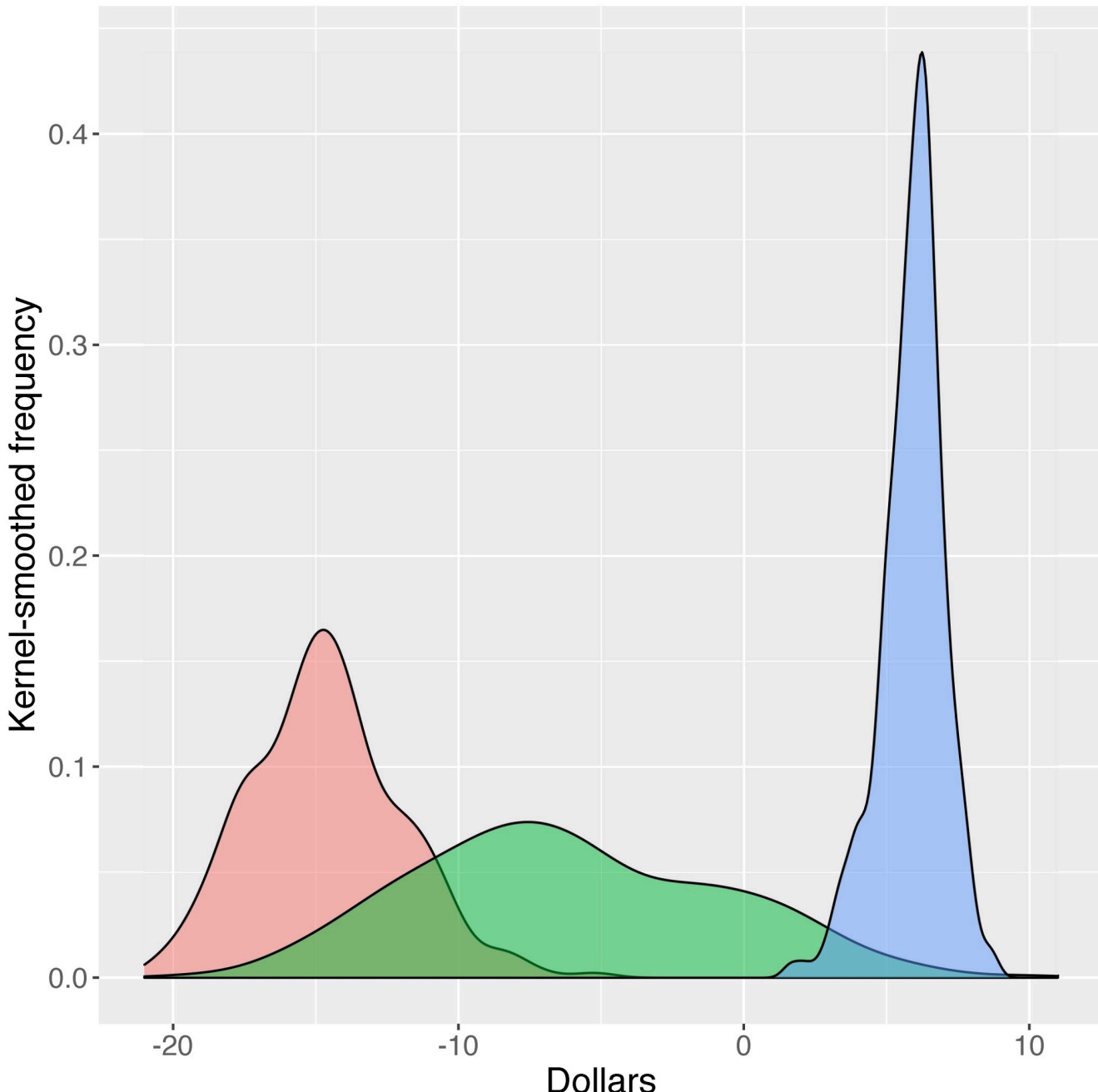

**Fig 5. Mean willingness to pay for PrEP among 236 young men who have sex with men in the U.S.: Additional options.** Red: Wait 4 weeks to start PrEP, compared to immediate start; Green: Wait 1 week to start PrEP, compared to immediate start; Blue: Communicate with provider online, compared to in-person visits.

insurances or enroll in a new insurance were significant detractors for the group of participants on parental insurance.

## Discussion

PrEP is fundamental to ending the HIV epidemic in the U.S. Use of PrEP among MSM who are at high risk for HIV is suboptimal, and we know less than we should about why that is the

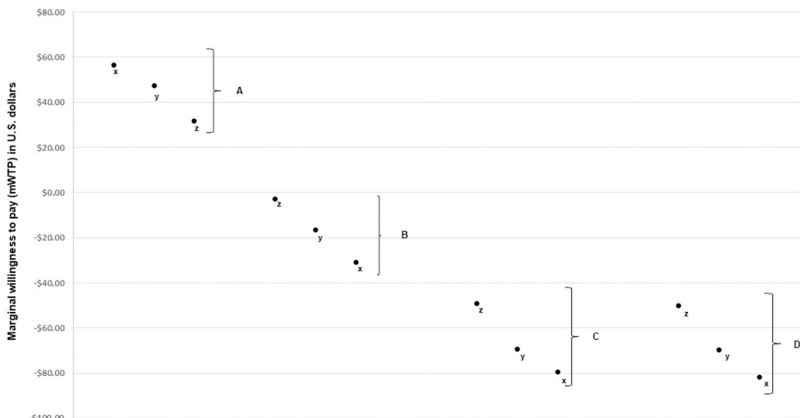

**Fig 6. Marginal willingness to pay for PrEP by insurance type for various insurance coverage scenarios among 236 young MSM.** A. PrEP is private (employers or family members cannot see benefits used); B. PrEP is not insurance covered; C. PrEP requires extra insurance; D. PrEP requires switch to a new insurance. Baseline insurance status. x. Participants on parental insurance; y. All participants, regardless of insurance status; z. Participants with employer-based exchange-based insurance.

case. After formative work identified affordability and insurance access as potential barriers to PrEP among young MSM, we created a DCE that probed specific preferences on these matters to better understand how to modify PrEP access to make it most attractive to this key group. Our online DCE that drew a racially diverse sample of young (18-30-year-old) MSM whose behaviors put them at high risk for HIV but who have not engaged in PrEP care. The results indicate that insurance coverage for PrEP is the most valuable of the attributes studied to our participants. Same-day access to PrEP and online PrEP options may also improve access. Stakeholders in our in-depth interviews indicated that they would not consider PrEP due to a perception that it would not be covered by their insurance. Utilizing mWTP analysis, our DCE indicates that young MSM are more willing to pay for PrEP that is covered by insurance and that having to take out a supplemental or new insurance was significantly and substantially less desirable. We believe these results reflect several realities that are notable, particularly in the U.S. health care system. First, out-of-pocket costs of health care in the U.S. is often astronomically high and unaffordable to a broad swath of the population [35,36]. Second, insurance complexity has been shown to detract from health care; lower health insurance literacy has been associated with lack of enrollment in insurance plans and lack of uptake of preventative services [37,38]. We postulate that an insurance-covered service is seen as more desirable and more likely to be affordable even if there is a co-pay associated with it. In contrast, the time and complexity associated with seeking a new or supplemental insurance may have made these options even less attractive than having to pay out-of-pocket. These results are entirely consistent with other studies showing lack of insurance as a barrier to PrEP care, even when individuals are given other stipends that theoretically should help offset costs [10]. The results are also consistent with a DCE that evaluated PrEP preferences for people who inject drugs in the U.S., and found insurance coverage to be among the most preferred features of PrEP [23]. It may be necessary to expand insurance coverage of PrEP specifically, and insurance coverage more broadly, in order to enhance uptake of PrEP, and offer of supplemental PrEP-specific insurance may not be enough to get around insurance barriers unless the ease of uptake is made very clear to those at risk. Previously cited data highlight that younger age groups are most likely to be uninsured and also face insurance instability, while moving from school to employment options; these employment options may be transitory and not offer insurance at

all. So, suboptimal uptake of PrEP among high risk young individuals is likely to remain problematic until better insurance coverage is regularly available for these groups.

While the ability to retain parental coverage through the Affordable Care Act was meant to mitigate the risk of young people being un- or under-insured, concerns about privacy may limit young MSM's use of PrEP. Our data demonstrate that participants were willing to pay significantly less for PrEP when their PrEP status might be revealed to employers or family members through insurance claims. In fact, after examining seven subgroups of our national sample, the most significant differences in preferences were found between the participants on parental insurance and those with other insurance types or no insurance. Participants with parental insurance appeared to be satisfied with the stability of the coverage (they were the least likely to be willing to take out a new or supplemental insurance for PrEP) but very concerned about privacy issues (they were the most willing to pay for insurance that is private). The ability for individual members to request privacy of their records, or enacting measures to bill up-front for all services, and make sexual health-related care not visible on estimation of benefits statements may reduce privacy concerns in this group [39].

Facilitating access to PrEP through rapid start and online PrEP programs may also be important steps towards enhancing PrEP appeal for high risk young MSM. Immediate access to PrEP was preferred to a one week or four-week wait, with a four week wait the least preferred alternative. Our data demonstrate an overall significant preference for online care with lab visits for testing, compared to in-person visits with a provider. Young people's lives, busy and mobile as they tend to be, may not fit well with standard, more slow-moving health care options. Additionally, many of the clients who are highest risk for HIV acquisition are young and healthy; moving their care out of a clinical space may offer them convenience and remove them from a sense of having a disease which needs to be managed. Some online PrEP programs have emerged—particularly as the COVID-19 epidemic in the U.S. has changed reimbursement for telemedicine visits. Nevertheless, there is a great need to evaluate their effectiveness, cost, and accessibility and to compare their performance to that of more standard clinic-based PrEP care. Of note, Hispanic participants and those who had high HIV risk due to alcohol and drug use demonstrated less preference for online care; these merit further exploration in other studies as well as adequate consideration in program design.

Our study has several unique advantages. To our knowledge, this is the first study that specifically examines PrEP access issues using a discrete choice experiment. Our participants represented a particularly high-risk group (young MSM who report recent condomless anal sex with non-monogamous partners) from broad geographical swath of the U.S., and included a substantial proportion of Hispanic men, who have increasing HIV incidence in recent years. While our sample size was relatively small, our participants were carefully screened to include those who were high risk and not on PrEP, many of the very people who we are most concerned about attracting to PrEP care from a public health standpoint. Our study also carefully screened for and omitted untrustworthy surveys (such as those possibly completed only for financial gain), likely enhancing the validity of our results [27,40]. In addition, social desirability bias may be somewhat mitigated by online surveys, allowing for better assessment of people's behaviors and preferences around sensitive needs areas such as PrEP.

There are also limitations of our study which are important to acknowledge. There was relatively low enrollment of Black MSM, who continue to face extraordinarily high risk of HIV acquisition and are under-represented in PrEP programs. Their voices remain critical in enhancing current PrEP options. Mistrust of a medical system that has often failed Black Americans, and structural barriers to healthcare particular to communities of color are not addressed by our DCE data, but were demonstrated to be barriers in the in-depth interviews.

Additionally, our recruitment approach and use of online surveys likely weighted our sample towards social media users, who may have different preferences compared to the larger YMSM population. The use of only two platforms (due to difficulty in engaging with other platforms) and non-randomized approach to sampling might have introduced other biases among social media users that are difficult to account for. Our DCE did not include options about which type of practice someone might prefer to receive PrEP care in (i.e. a sexual health clinic, a primary care clinic, or student health center); preference might exist around such locations, or perspective clients may perceive access barriers based on not knowing where to access PrEP, which was also not addressed in this DCE format. Finally, any survey cannot fully predict behaviors, though discrete choice experiments appear to better allow for assessment of priorities compared to many other types of survey research.

While our study was specifically designed to assess the preferences of young MSM, age groups less than 18 were not included. Psychological aspects of health care engagement in youth were also not able to be explored in this study, including feelings of invincibility associated with youth, and difficulties in future-oriented care. It was notable that a substantial proportion of our participants self-assessed as being at low risk for HIV despite their arguably high risk based on the objective assessment of available data for the U.S. and their reported behavior.

## Conclusions

Our national convenience sample of young MSM who use social media, have never used PrEP, and have recently engaged in condomless anal sex with casual or non-monogamous partner(s) demonstrates that concerns about insurance coverage and insurance privacy significantly influence the likelihood of reported PrEP acceptability. Improved national protocols to guarantee PrEP coverage and users' privacy along with faster access to PrEP care, and online PrEP care may substantially enhance uptake of this critical service for youth and young adult MSM who remain at particularly high risk for HIV.

## Supporting information

**S1 File. Code screen & closed-ended questionnaire.**
(DOCX)

**S2 File. Data DiCE4PrEP de-identified.**
(XLSX)

**S3 File. DiCE supplementary material 19 May 21.**
(DOCX)

**S4 File. Relevant transcripts DiCE4PrEP.**
(DOCX)

## Acknowledgments

We appreciate all of our participants and our in-depth interviewers, Michael Branch, Timothy Bryant and Aaron Hoy.

## Author Contributions

**Conceptualization:** Elizabeth A. Asiago-Reddy, James McMahon, Andrew S. London.

**Data curation:** Elizabeth A. Asiago-Reddy.

**Formal analysis:** Elizabeth A. Asiago-Reddy, John McPeak, Riccardo Scarpa, Amy Braksmajer, Andrew S. London.

**Funding acquisition:** Elizabeth A. Asiago-Reddy.

**Investigation:** Elizabeth A. Asiago-Reddy, Amy Braksmajer.

**Methodology:** Elizabeth A. Asiago-Reddy, John McPeak, Riccardo Scarpa, Amy Braksmajer, Andrew S. London.

**Project administration:** Elizabeth A. Asiago-Reddy, Amy Braksmajer, Nicola Ruszkowski, Andrew S. London.

**Supervision:** Elizabeth A. Asiago-Reddy, James McMahon, Andrew S. London.

**Validation:** Elizabeth A. Asiago-Reddy, Riccardo Scarpa.

**Visualization:** Riccardo Scarpa.

**Writing – original draft:** Elizabeth A. Asiago-Reddy, Andrew S. London.

**Writing – review & editing:** Elizabeth A. Asiago-Reddy, John McPeak, Riccardo Scarpa, Amy Braksmajer, Nicola Ruszkowski, James McMahon, Andrew S. London.

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
