## [Decision Letter · Decision Letter 0]

31 Mar 2021

PONE-D-20-40977

Perceived access to PrEP as a Critical Step in Engagement: A Qualitative Analysis and Discrete Choice Experiment among Young Men Who Have Sex With Men

PLOS ONE

Dear Dr. Asiago-Reddy,

Thank you for submitting your manuscript to PLOS ONE. After careful consideration, we feel that it has merit but does not fully meet PLOS ONE’s publication criteria as it currently stands. Therefore, we invite you to submit a revised version of the manuscript that addresses the points raised during the review process.

We look forward to receiving your revised manuscript.

Kind regards,

Rupa R. Patel, MD

Academic Editor

PLOS ONE

Journal Requirements:

2. Please include additional information regarding the survey or questionnaire used in the study and ensure that you have provided sufficient details that others could replicate the analyses. For instance, if you developed a questionnaire as part of this study and it is not under a copyright more restrictive than CC-BY, please include a copy, in both the original language and English, as Supporting Information."

3. Please clarify the date range for which you collected data. The current manuscript states "From July to January, 2018" which seems like an error."

4. Please list the name and version of any software package used for statistical analysis, alongside any relevant references. For more information on PLOS ONE's expectations for statistical reporting, please see https://journals.plos.org/plosone/s/submission-guidelines.#loc-statistical-reporting.

6. Please include a caption for figures 3 and 4.

Additional Editor Comments (if provided):

The authors present an innovative study design. Major revisions are required for further clarification for the audience.

Abstract

-Please include the number of in-depth interviews and the number of online surveys conducted within the methods section. In the results section of the abstract, please remove the online survey sample size.

-In line 55, a more accurate term for insurance enrollment could be “enroll in” new insurance.

Reviewer #1 presents with important points regarding the Intro and Methods sections.

-Insurance is not a modifiable category as Reviewer #1 presents. Please revise these sections.

-DCE sampling has limitations and the authors should incorporate this in the manuscript

-Reviewer #1 brings up a good point regarding the subgroup analyses in the results section.

-Please comment on the high behavioral risk categorization that the authors discussed and as reviewer #1 brings up as a limitation.

-As suggested by Reviewer #2, there needs to be more detail on the methods and the definition of grounded theory.

-NVivo needs to be revised for the spelling.

-In line 246, please provide a reference.

Results

-Table 1 requires a demographic breakdown of participants.

-Clarification of the definition of PrEP access is needed.

-Clarification on ranking is required.

-In line 297 regarding ACA, please provide a reference.

Discussion

-Reviewer #1 indicates that the results are overstated; please revise the discussion section in response to Reviewer #1's comments. Please comment on the study limitations regarding not having a national representative sample for the in-depth qualitative interviews and their influence on the online survey.

-In line 521, the authors should highlight study limitations instead of disadvantages.

Figures

-The Figure text is not legible.

Reviewers' comments:

Reviewer's Responses to Questions

**Comments to the Author**

1. Is the manuscript technically sound, and do the data support the conclusions?

Reviewer #1: Yes

Reviewer #2: Yes

2. Has the statistical analysis been performed appropriately and rigorously? 

Reviewer #1: Yes

Reviewer #2: Yes

3. Have the authors made all data underlying the findings in their manuscript fully available?

Reviewer #1: No

Reviewer #2: Yes

4. Is the manuscript presented in an intelligible fashion and written in standard English?

Reviewer #1: Yes

Reviewer #2: Yes

5. Review Comments to the Author

Reviewer #1: The authors have conducted an elegant mixed-methods investigation that deepens our understanding of PrEP uptake among young MSM. The use of the discrete choice experiment methodology is novel, and the manuscript should inspire other investigators to employ this technique more widely in HIV prevention research. The compelling findings the authors present have important implications for the development of PrEP programs serving young MSM. The manuscript is very well-written and there are only a few places that needed strengthening. Mainly, the authors should provide additional details about the qualitative analysis methods used in the first component of the study, and about the recruitment methods for both components of the study.

Lines 119-139 describing the in-depth interviews should be strengthened. The authors did not describe how participants were recruited for the in-depth interviews, nor did they include full demographic information on the group of interview participants. The authors should add these demographic data as a table or in the main text. The authors should also consider describing relevant demographic characteristics of the interviewers (i.e. were they also young MSM or otherwise peers or near peers of the interview participants?) and findings from any reflexive activities that were conducted during the interview data collection cycle. If the authors conducted a quantitative assessment of inter-rater reliability between interviewers, they should include that data in this section as well. Additionally, the description of the qualitative analytic techniques the authors used warrants a more detailed description. The authors can do this by providing a short definition of both grounded theory and of inductive methods, as well as describing how these methods were used throughout the interview cycle.

Line 157-158 The authors should provide information about the types of social media applications that were used to recruit participants. Were these traditional social media sites like Instagram and Facebook or were they hookup/dating apps like Grindr and Scruff (or a combination of the two). Delineating this can help the reader better understand the sampling strategy.

Line 532 very minor typo- should read “which was also not addressed.”

Reviewer #2: This purpose of this paper was to assess preferences for PrEP access in the United States among YMSM through discrete choice experiments informed by in-depth interview findings. Overall, this was a very interesting and novel approach to assess how to improve PrEP uptake among a population with higher HIV incidence; however, there are design and methodologic concerns that present major barriers to the generalizability of the findings. I have the following suggested edits and comments for the authors:

Introduction

Overall, the introduction is well written. However, a stronger argument needs to be made for the structural barrier attributes included in this analysis, as insurance status is not easily “modifiable”. This can be acknowledge/justified more strongly in the introduction (as an introduction to the concepts) or in the methods. While insurance status can change, it would more likely need adjustments in policy versus being modifiable on an individual-level. To inform future research, this work identified a clear determinant for PrEP uptake, but interventions targeting insurance coverage may be limited (especially in states that have not expanded Medicaid).

Methods

Major concerns are related to the in-depth interviews being conducted with participants from Upstate New York and the DCE surveys being distributed nationally. If attributes selected were partially informed by the qualitative findings (esp. structural barriers), this leads to bias in the attributes selected for DCE in the survey because of the IDI participants contextual barriers related to the region in which they live as well as factors related to their identity/race/ethnicity (which is not provided). If additional attributes were selected based on literature review, then this should be clearly outlined in the methods section as to which attributes were informed by the researchers’ findings versus by the literature review. Also, the authors should describe how they decided on which PrEP characteristics to have participants rank for in-depth interviews?

Additionally, the authors do not describe if they had a target sample size based on the number of levels within attributes as well as choice sets created. Also, I would have considered a geographic subgroup analysis, based on looking at structural barriers. Insurance coverage and acceptance of costs may differ based on the region in which YMSM live.

Page 5, line 105 – “relative importance of PrEP attributes”. I would recommend delineating attributes related to PrEP itself and service delivery attributes to help with clarity.

Page 13, line 241 – Why is the “high behavioral risk” group > 10 anal sex partners in the last 6 months while never using condoms? I understand wanting to show high risk, but based on prevalence and CDC guidelines, I am unsure if such a high number is necessary to illustrate risk.

Results

Major concerns in the results section for the in-depth interviews, was not describing demographics for participants in text or in a table. Also, it was very hard to read Figure 1-3. I would recommend the authors make the text more legible and look at different color scales.

Page 15, Line 307 – “This example reveals how stigma and privacy concerns, potentially related to being…” This seems like a sub-theme related, but distinct from privacy concerns with parents. The concern that participant will be judged for being on PrEP by “people” may support the need for other interventions addressing stigma among late adolescents who are not able to realistically obtain privacy in billing for PrEP when on their parents insurance plan.

Page 16, Line 326 – The label “Prioritization of PrEP Elements” should potentially be changed to attributes for clarity sake.

Page 17 – Overall with the ranking process, I had a few questions? Again, how were attributes included for ranking selected, was this based on literature review and if so it should be clearly stated? How was access defined by the authors for participants, because it could have several meanings? This is made even more confusing the sentence, “Assuming high effectiveness, access issues – cost (insurance coverage, privacy) and ease of access – were highly important, since from the standpoint of participants, some the specifics of use did not really matter if you could not get it.” This sentence makes it appear that cost was included in access.

Discussion

Overall, the major findings outlined in the discussion are stated too strongly in that YMSM may not be pursuing PrEP due to costs and insurance coverage. I think the authors need to better present the context when drawing major conclusions from the results. This paper largely focused on structural barriers to PrEP access, which are unique and need to be addressed. However, stigma and other individual- and community-level barriers are also important in improving PrEP access. Additionally, this survey was done with a sample not entirely reflective of current HIV epidemiology in that it did not have a large enough representation of BMSM and persons living in the South, with bias in it being conducted over an app which likely skewed results in preference for on-line PrEP services. In saying this, the results still add to the literature in the type of analysis done and the impact it may have on future intervention development. This is one piece of a complex puzzle when trying to improve PrEP uptake among YMSM.

Page 23, Line 451 – “Results from in-depth interviews that informed the design of the DCE provided key explanatory evidence that concerns about affordability supersede concerns about mode of administration, frequency of administration, side effects, and stigma associated with taking PrEP.” This is an example of drawing conclusions not necessarily supported by the results. First, the IDIs were done to inform DCE and, therefore, cannot be explanatory but are exploratory. Also, I am not sure how based on IDI findings the authors have come to the conclusion that affordability supersedes the factors listed above. That is not the goal of qualitative research and these factors were not included in the DCE analyses.

I would recommend rewriting the discussion section with these overarching comments in mind.

In all, this paper was well written and provides a unique analysis to better understand how to improve PrEP access among YMSM. I believe that with the major revisions suggested above, it will be a great addition to the literature.

6. PLOS authors have the option to publish the peer review history of their article (what does this mean?). If published, this will include your full peer review and any attached files.

Reviewer #1: **Yes: **Justin C. Smith

Reviewer #2: No

---

## [Author Response · Author response to Decision Letter 0]

30 Jul 2021

The detailed response to reviewers is contained in the document attached to the manuscript files, wherein items are addressed one by one.

---

## [Editor Report · Decision Letter 1]

30 Jul 2021

PONE-D-20-40977R1

Perceived access to PrEP as a Critical Step in Engagement: A Qualitative Analysis and Discrete Choice Experiment among Young Men Who Have Sex With Men

Dear Dr. Asiago-Reddy,

The above-mentioned manuscript has now been withdrawn from the review process at PLOS ONE.

If this was a mistake, please do reach out to us at plosone@plos.org. Otherwise, no further action is needed from you at this point.

Sincerely,

PLOS ONE

---

## [Author Response · Author response to Decision Letter 1]

7 Sep 2021

Thank you very much for the opportunity to revise the manuscript. A letter containing all of the requested changes has been included in the revision. All of the edits requested to the format of the manuscript have been applied to the best of my understanding.

---

## [Editor Report · Decision Letter 2]

30 Sep 2021

Perceived access to PrEP as a Critical Step in Engagement: A Qualitative Analysis and Discrete Choice Experiment among Young Men Who Have Sex With Men

PONE-D-20-40977R2

Dear Dr. Asiago-Reddy,

We’re pleased to inform you that your manuscript has been judged scientifically suitable for publication and will be formally accepted for publication once it meets all outstanding technical requirements.

Kind regards,

Rupa R. Patel, MD

Academic Editor

PLOS ONE

---

## [Editor Report · Acceptance letter]

15 Oct 2021

PONE-D-20-40977R2 

Perceived Access to PrEP as a Critical Step in Engagement: A Qualitative Analysis and Discrete Choice Experiment among Young Men Who Have Sex with Men 

Dear Dr. Asiago-Reddy:

I'm pleased to inform you that your manuscript has been deemed suitable for publication in PLOS ONE. Congratulations! Your manuscript is now with our production department. 

Kind regards, 

on behalf of

Dr. Rupa R. Patel 

Academic Editor

PLOS ONE